# Multicultural Identity Integration versus Compartmentalization as Predictors of Subjective Well-Being for Third Culture Kids: The Mediational Role of Self-Concept Consistency and Self-Efficacy

**DOI:** 10.3390/ijerph20053880

**Published:** 2023-02-22

**Authors:** Magdalena Mosanya, Anna Kwiatkowska

**Affiliations:** 1Institute of Psychology, Polish Academy of Sciences, Jaracza 1, 00-378 Warsaw, Poland; 2Psychology Department, Dubai Campus, Middlesex University London, Knowledge Park, Bldg. 16, Dubai P.O. Box 500697, United Arab Emirates

**Keywords:** third culture kids, multicultural identity, well-being, self-consistency, self-efficacy

## Abstract

Globalization has resulted in an exponential increase in the number of Third Culture Kids (TCKs), defined as being raised in a culture other than that of their parents (or the passport country) and meaningfully interacting with different cultures. Inconsistencies regarding the effect of multicultural and transient experiences on well-being exist in the psychological literature. We aimed to reveal associations between multicultural identity configurations (integration, categorization, compartmentalization) and well-being with the mediating role of self-concept consistency and self-efficacy. Participants (*n* = 399, *M* = 21.2 years) were students at an international university in the United Arab Emirates. We used the Multicultural Identity Integration Scale, the Berne Questionnaire of Subjective Well-Being, the General Self-Efficacy Scale, and the Self-Consistency Subscale from the Self-Construal Scale. The findings suggest that not merely exposure to diversity but also internal integration versus identity compartmentalization moderate the well-being of TCKs. We explained such mechanisms via partial mediation of self-consistency and self-efficacy. Our study contributed to a better understanding of the TCKs’ identity paradigm and pointed to multicultural identity integration as vital to TCKs’ well-being via its effect on self-consistency and self-efficacy. Conversely, identity compartmentalization decreased well-being via a reduction in the sense of self-consistency.

## 1. Introduction

### 1.1. Background

Globalization processes have prompted a rise of global citizens, so-called Third Culture Kids (TCKs) or global nomads, who move around the world following their parents and experiencing migrations during developmental years [1,2]. The exposure to diverse cultural paradigms on a never-seen-before scale has impacted the traditional understanding of self-concept and identity development. Crucial deviations from standard models have provoked a new wave of research on culturally complex identities [3,4]. However, the topic concerning new identities and how individuals deal with internal cultural diversity has yet to be sufficiently explored. Although TCKs’ cultural and social identities are generating more and more attention from researchers, there need to be more studies on the other aspects of TCKs’ self-concept, i.e., the sense of self-consistency or self-efficacy. Furthermore, recent research pointed to multicultural identity configurations (understood as ways of dealing with internal cultural diversity) as predictors of well-being for TCKs [3]. However, the mechanism behind such an effect still needs to be explored. The present study aimed to fill this gap, proposing intermediary effects on self-consistency and self-efficacy. 

### 1.2. Literature Review

#### 1.2.1. Third Culture Kids

Third culture kids (TCKs) are defined as raised in a culture other than that of their parents (or the culture of the country given on their passport) for a significant part of their early developmental years [1]. TCKs spend developmental years living overseas, meaningfully interacting with two or more cultural environments, which significantly influences their sense of identity shaped during adolescence [1,2]. TCKs further blend different cultural frames within themselves and might have the adaptability to feel at home everywhere, treating the world as a “global village”. Multicultural exposure has, therefore, multiple advantages. TCKs are recognized for their multilingualism, global leadership skills, global mindset, and intercultural sensitivity [2,5,6]. TCK individuals increasingly identify themselves as a separate group with a shared identity [2,7] and create organizations (e.g., Mu Kappa) to promote complex identities and provide support to people with multicultural and “nomadic” experiences. Alternatively, TCKs exposed to multiple cultural paradigms may experience difficulty finding groups to which they feel a sense of belonging [1,5,8,9]. TCKs need to negotiate their identities in early developmental years [10], which might jeopardize consistent identity formation and provoke fragmentation.

Due to accelerated economic growth, the United Arab Emirates, a Gulf country and hub for international business, brought together many expatriate workers to settle within its borders. As a result, the social environment of the UAE is truly multicultural. Furthermore, within the country, diversity and integration rather than assimilation are promoted. As young citizens can be predominantly characterized as TCKs [11,12], the UAE constitute an exciting center for well-being-related TCKs research.

#### 1.2.2. Self-Concept and Identity Forms

Within the identity literature, the term self (self-concept) is applied in reference to a mental representation of oneself (e.g., [13,14,15,16,17]). Self-concept reflects self-schema, “a collection of at least semi-related and highly domain-specific knowledge structures” (p. 182, [18]). The function of the self-concept is to configure the information and regulate intentional behavior [19]. Hence, the self-concept can be understood as essential knowledge about the individual as part of a specific cultural/social environment (e.g., the self-concept constructed in one setting would be different from the self-concept created in another).

Furthermore, there is no consensus on whether self and identity are distinct or overlapping concepts. Some psychologists use these terms interchangeably [15], while for others, these words have differentiated meanings [20]. In this paper, to avoid possible confusion, we focused on identity understood as a part of the self-concept, i.e., as a specific subset of self-construals (forms of identity) which are relatively central as opposed to peripheral, essential as opposed to marginal, and substantial as opposed to nonmaterial [21,22]. The self-concept content may include all possible characteristics of a person or a group, but only some may be attributed to personal or social identity.

Identity is understood as a subjectively experienced concept of oneself as a person [23]. The identity process theory (IPT) [23,24,25] has brought a novel, dynamic perspective on self-construction, highlighting the socio-psychological processes underlying identity creation. Such a viewpoint encompasses a constructivist paradigm, within which identity is characterized as a multifaceted notion that continuously undergoes transformations based on interactions with changing contexts. Therefore, identity may take fluid and flexible forms. Such an approach seems particularly relevant when discussing new paradigms of TCKs and their identities.

Vignoles [25] further explained that identity indicates an answer to the question: *Who am I?* This question may appear in two forms: *Who am I as an individual?,* and *Who am I as a social being?* Thus, identity may be defined at two levels: individual and social (in line with the social identity theory (SIT)) [26]. Consequently, *individual* or *personal identity* refers to personal differences and attributes. *Social* or *collective (including cultural) identity* refers to identification with groups and social categories. *Cultural identity* is derived from membership and self-identification with a cultural group [26]. A cultural group may consist of people of common ancestry (e.g., an ethnic identity, [27]) and people sharing common values and practices. Such a view on *cultural identity* goes beyond nationality or ethnicity [28]. According to Vignoles [25], a characteristic becomes part of identity only if it is infused with a personal and social meaning. Hence, the identity reflects the most salient aspects of one’s self-concept, which is culturally shaped. 

#### 1.2.3. Cultural Context and Identity 

The adaptive function of culture is seen as a customary way of acting, feeling, and thinking chosen by society from an infinite number and variety of possible ways of being. Cultural systems incentivize individuals to function in a specific frame. This particular frame influences how people see themselves in relation to others, so-called self-construals defined as different patterns in how the “self” is constructed (dependent vs. independent) [29]. Identity construction occurs through the acquisition of specific properties by self-construals based on an ongoing complex interplay of cognitive, affective, and social interaction processes [23,24]. Vignoles et al. [30] explored the conceptualization and measurement of traditional views on self-construals and proposed to see the constructs of selfhood as multidimensional, with different ways of being independent and interdependent. According to Vignoles [31], people are motivated to construct identities that allow them (among others) to have a feeling of being the same over time and across situations despite significant life changes (the consistency) and to feel competent and capable of influencing their environment (the efficacy). A sense of consistency, a sameness across situations, is experienced diversely across cultures [24]. Similarly, though a universal construct, self-efficacy is impacted by enculturation processes and originates from social and cultural practices. So, for multicultural TCKs, these two dimensions of self-concept would also have a specific presentation and function. Importantly self-concept dimensions have widespread implications for people’s psychological and social experiences. At the individual level, they have been seen as predictors of cognition and motivation [32,33] and mediators in the effect culture has on emotions and cognitive processes [34,35].

#### 1.2.4. Self-Consistency, Self-Efficacy and TCKs

Some dimensions of self-concept might be more crucial to third-culture individuals due to TCKs’ specific, transient life experiences. The exposure to different cultural contexts combined with high mobility and frequent transitions from place to place may threaten TCKs’ sense of consistency, understood as being the same in time and across situations. Furthermore, being an object of decisions made by somebody else (parents, institutions) concerning where to live, or study, may diminish one’s sense of agency and frustrate self-efficacy. Hence, the self-concept-related dimensions of self-consistency and self-efficacy may have an ambivalent presentation for TCKs, discussed below. 

Self-concept consistency or self-consistency is considered a defining feature of identity [10,36] and indicates one’s perceived consistency (sameness) across situations and time [25,30]. In his developmental model, Erikson [10] pointed to the adolescent years as essential for developing identity clarity versus confusion. Erikson further highlighted that identity coherence and unity were based on solid connections to precise socio-cultural paradigms, values, and beliefs. These might be jeopardized in the case of third-culture individuals. Furthermore, studies suggest a close and dynamic relationship between the continuity and consistency of the narration of culturally significant memories and identity development [37]. The defining feature of TCKs’ lives is high mobility [1]. Children and young people who follow their parents overseas, experience an endless pattern of relocations and changes. These may break TCKs’ developmental trajectories by setting new goals to achieve, new values to respect, and new rules to follow. Hence, TCKs may experience incoherence across situations and the unpredictability of the future [38]. Furthermore, TCKs’ temporary and situational fragmentation might constitute an issue for functioning and self-evaluation [37]. 

Self-efficacy appears to be significant to identity and the general functioning of multicultural individuals, though its role might be intermediary [23]. An individual’s sense of self-efficacy determines whether one sets goals and acts on them. Bandura [39] theorized that self-efficacy is a context-specific judgment about one’s ability. In a cross-cultural context, Hoersting and Jenkins [8] evidenced that self-efficacy was buffering a negative impact of recurrent relocations on TCK adolescents’ coping and adjustment. Intercultural competency further increases bicultural individuals’ functioning [40]. For TCKs, self-efficacy might be, on one side, impaired due to a lack of control over multiple relocations and inevitable change. On the other hand, self-efficacy might be enhanced due to TCKs’ vast cross-cultural competencies, including a global mindset [2]. 

#### 1.2.5. Multicultural Identity Configurations

Exposure to cultural diversity constitutes a challenge to the psychosocial development of TCKs, particularly to establishing a secure and coherent cultural identity. In cultural identity construction, TCK individuals may experience conflicting demands and expectations from different cultural groups encountered in their mobile lives, resulting in difficulties in achieving a coherent cultural identity. Consequently, for third culture kids, the discourse on multiculturalism is shifting from external influence, i.e., acculturation, to the internal fusion/hybridization of their cultural selves. Hybrid identities are fluid and transformative and do not fall into traditional cultural categories [41]. 

To encompass cultural pluralism, multicultural individuals ought to engage in specific strategies facilitating their identity building. The cognitive-developmental model of social identity integration (CDSMI) [42] accounts for the different ways that multiculturals cognitively configure their many cultural identities within their overall identity. Research has identified three types of identity configurations: categorization, compartmentalization, and integration [4,42,43]. The categorization configuration implies identifying with one cultural group, seeing one identity as predominant, and excluding other identities from the self. When endorsing categorization, differences between a chosen group and other groups are likely to be salient [43]. In contrast, the compartmentalization configuration allows an individual to endorse multiple identities, but they are kept separate from each other. These identities are context-dependent and activated depending on the social context. Although an individual may identify with many cultures, identities based on these cultures are not linked to one another within the self [43]. The third configuration—integration—occurs when individuals feel that they endorse belonging to different cultural groups. Thus, multiple identities are organized within the self to be equally essential and form one coherent supra-identity. The differences are seen as complementary and enriching to oneself. Integration may enable individuals to establish context-independent superordinate identity encompassing multiple influences [4] that cannot be reduced to the sum of its constituent identities [43,44]. The integrated multicultural identity has been linked to a global mindset in Mosanya and Kwiatkowska’s [3] study, suggesting it may involve cosmopolitan and supra-cultural aspects reflecting the hybridization of identity [45]. The following quote from an interview with TCK supports such thesis “*I am a global citizen (…) I have an identity just not the national one*” (p. 23, [38]).

Aspects of self-concept and cultural identity are certainly reciprocally connected, though existing research has focused mainly on ethnic or national identity, possibly as a proxy for cultural identity [27,46,47]. The present study focuses on the relationship between multicultural identity configurations and self-concept dimensions of self-consistency and self-efficacy vulnerable for TCKs, based on the following premises. Usborne and de la Sablonniere [47] concluded that the clarity and consistency of cultural identity determine the clarity of self-concept and, consequently, well-being. The divergent selves with contradictory meanings proved challenging to identity integrity, causing stress and a loss of self-efficacy, and impaired well-being. Achieving an optimal sense of consistent and efficacious identity is also possible for TCKs through another mechanism, namely by identification with other TCKs. Identification as TCKs reflective of integrated identity configuration could fulfil the motive for consistency via attributions of the consistent TCK experience of transience and mobility. Some evidence can be found in qualitative research where individuals self-refer as TCKs, e.g., “me as ti-si-kayz” and “my TCK tribe” [7].

#### 1.2.6. Well-Being and TCKs’ Identity 

One of the most significant life aims is subjectively experienced well-being (SWB), a primary interest of positive psychology, explained as a study on human flourishing. SWB is defined as a person’s cognitive and affective evaluations of life, a combination of feeling good and functioning well [48]. SWB is a multidimensional notion understood in various ways, e.g., as life satisfaction, positive affect, optimal functioning or happiness [49], sense of control over one’s life [50], and a blend of psychological positivity, physiological health and ill-being [51,52]. Life satisfaction can further be explained as a state of achievement linked to positive affect and the ability to deal with life circumstances [23,53]. Life satisfaction has been a predictor of individual and societal welfare [54].

Previously, studies concerning multicultural individuals have claimed that frequent geographical relocations had a long-lasting negative effect on self-concept clarity and general functioning [8,55]. In particular, the mental health of TCKs was seen as fragile and requiring support. A recent study conducted in the UAE on a sample of TCKs evidenced that nearly 30% reported moderate to severe depressive symptoms [11]. However, alternative research points to the cultural identity configurations as significant factors in multicultural individuals’ well-being, with integration identified as the positive predictor of life satisfaction [3]. A coherent self-concept, formed based on the integration of diverse cultural paradigms, has been linked to better mental health and higher self-esteem [27], both predictors of well-being. Moreover, a sense of self-consistency and self-efficacy are associated with positive emotions and enhanced well-being, whereas if frustrated, these dimensions of self may be related to ill-being [23,56]. Furthermore, the narrative continuity of identity and internal integrity relate to positive affect and well-being [23,56]. Likewise, a sense of self-efficacy for individuals with multiple cultural selves facilitates the socio-cultural adaptation process and supports well-being [57]. 

### 1.3. Aims and Hypotheses

In a novel way, the present study linked the established theories of multicultural identity configurations [4,42] with self-consistency and self-efficacy with multifold objectives [8,10,30,58]. Firstly, we explored whether significant relationships exist between configurations of cultural identity (integration, categorization, compartmentalization) and a sense of self-consistency, self-efficacy and well-being [3]. We further investigated if there were intermediary effects among multicultural identity configurations in their predictive effects on well-being with self-consistency and self-efficacy in the role of mediators. Figure 1 provides a visualization of the following hypotheses:

**H1.** 
*Multicultural identity integration positively predicts well-being (H1a), and such a relationship is mediated by increased self-consistency (H1b, c) and self-efficacy (H1d, e);*


**H2.** 
*Multicultural identity compartmentalization negatively predicts well-being (H2a), and such an effect is mediated by compartmentalization’s negative effect on self-consistency (H2b, c) and self-efficacy (H2d, e);*


**H3.** 
*Multicultural identity categorization negatively predicts well-being (H3a), and such an effect is mediated by the level of self-consistency (H3b, c) and self-efficacy (H3d, e).*


## 2. Materials and Methods

### 2.1. Participants

The sample (*n* = 399) consisted of 296 females (74%) and 103 males (26%) with a mean age of 21.2 (*SD* = 3.5, Mode = 19, Range 18–43). Third culture individuals were considered for this study after they checked “yes” for the given definition: *Check YES if you have been raised in a culture other than that of your parents (or the culture of the country given on your passport) for a significant part (more than one year) of early developmental years 6*–*18* [1]. All participants were students at an international university in the UAE. Participants were from non-Western countries currently residing in the UAE, with 205 (51%) being Indian passport holders, followed by Pakistani 20 (5%), Arabs 20 (5%), Filipinos 18 (4.5%), and others. They reported being influenced by between 3–7 cultures (*M* = 3; *SD* = 3.5; Mode = 3); Most of them spoke, on average, three languages (Mode = 3, Range 1–6). Their religions were Muslims 40%, Hindu/Buddhist 22%, Christians 21%, atheists 8%, and others. All participants were fluent in English, a requirement for university admission.

### 2.2. Materials

All questionnaires were administered in English in their original versions, which are universally accessible. 

Sense of self-concept consistency was measured with the Self-Consistency Subscale (6 items, *α* = 0.80) from Vignoles et al. [30] consisting of the Self-Construal Scale, rated on a 7-point Likert scale of 1 (not at all) to 7 (exactly). Item sample: “*You behave the same way at home and in public*”. SCS has demonstrated reliability in cross-cultural studies [30].

Sense of self-efficacy was assessed with the General Self-Efficacy Scale [59] by incorporating ratings on a 7-point Likert scale of 1 (not at all) to 7 (exactly). It was a 10-item questionnaire (*α* = 0.88) with an item sample: “*I can always manage to solve difficult problems if I try hard enough.*” The GSES has shown reliability in past studies on predictors of life satisfaction with students of non-Western origin (e.g., [60]). 

The configurations of multicultural identities were determined by the Multicultural Identity Integration Scale (MULTIIS; [4]). It consisted of 22 items scored on a 7-point Likert scale of 1 (not at all) to 7 (exactly). MULTIIS contained three subscales: categorization (5 items, *α* = 0.75), item sample: “*One of my cultures is more relevant in defining who I am than the others.*”; compartmentalization (9 items, *α* = 0.80), item sample: *“I identify with one of my cultures at a time*.”; and integration (8 items, *α* = 0.82), item sample: “*I have an identity that includes all my different cultural identities*”. Confirmatory Factor Analysis (CFA) was performed to assess the three-factor structure of the MULTIIS. The CFA model provided acceptable fit to the data: *χ*^2^ = 398.41; *df* = 196; CMIN/*df* = 2.03; RMSEA = 0.051 [90% CI = 0.044–0.060]; CFI = 0.924. The MULTIIS scale has been employed in an exploratory study on female TCKs [3] and has shown a three-factor structure and reliability of subscales.

The Berne Questionnaire of Subjective Well-Being [61] measured subjective well-being. The scale included 39 items, rated on a 7-point Likert scale of 1 (not at all) to 7 (exactly). Item samples: life satisfaction: “*I am content with the way my life plans are being realised*”, and ill-being (R) “*I find my life uninteresting*”. We have used the total score of life satisfaction and reversed ill-being items (*α* = 0.91). The scale showed reliability in the cross-cultural assessment of adolescents [52].

### 2.3. Procedure

The study was not preregistered. Data were collected online. To assure anonymity and data confidentiality, the link to the study was posted on groups and platforms for international students in the UAE. Respondents consented to participation after being informed about the study objectives, non-paid participation, anonymity, confidentiality, and withdrawal rights. The contribution was voluntary, and participants were given an email to the researchers and asked to insert their initials only. Any communication should have mentioned this if they wished to withdraw from the study. The data were encrypted for safe storage.

### 2.4. Analytical Approach

It was a questionnaire-based and cross-sectional study. We employed Pearson’s correlation coefficient and the hierarchical multiple regression analysis. The mediation analyses were performed using Model 4 of PROCESS Macro [62]. We proposed a complex model; hence a parallel mediation, which included more than one mediator, was deemed most appropriate [62].

## 3. Results

### 3.1. Correlational Analyses

All variables were normally distributed with skewness coefficients and kurtosis between +1 and −1. Descriptive statistics of all scales are presented in Table 1. Pearson’s correlation coefficient analyses (Table 1) revealed significant associations between variables pairwise. Self-consistency was positively associated with integration and categorization. Self-efficacy was related positively only to integration while negatively to compartmentalization. Well-being was positively significantly associated with self-consistency and self-efficacy and multicultural identity integration but negatively with compartmentalization. There was no association between categorization and well-being.

### 3.2. Hypotheses Verification

#### 3.2.1. Direct Effects

A hierarchical multiple regression analysis was performed (Cook’s D values < 1) to investigate the influence of the factors related to social and cultural identity on well-being (Table 2). The predictors were grouped into two models: 1. Multicultural identity configurations; 2: Self-consistence and self-efficacy. Model 1 was a good fit for data (*p* < 0.005) and explained 12% of the variance in well-being with integration and compartmentalization as significant predictors. Model 2 was a better fit for the data. All predictors accounted for 41% of the variation in well-being, with integration, self-consistency, and self-efficacy being positive and compartmentalization being a negative predictor of well-being. Post-hoc power analysis for five predictors revealed a good power of detecting an effect (power = 0.99; *n* = 399; *R*^2^ = 0.41; *p* = 0.05).

#### 3.2.2. Mediation Analyses

After confirming the existence of pairwise correlations between variables and direct effects for integration and compartmentalization on well-being, the mediation models were evaluated with path c indicating the direct effect of IV on DV and path c’ representing effects with mediators included in the model. Paths a, b, c, and d show the necessary assumption for the mediation model. 

##### Hypothesis 1 (a, b, c, d, e)

A parallel mediation model analysis was used to investigate hypotheses that self-consistency and self-efficacy mediate the positive effect of integration on well-being (Figure 2). Results indicated a positive and significant total effect of integration on well-being supporting H1a. Integration was a significant predictor of self-consistency, *β* = 0.14, *SE* = 0.06, 95%CI [0.01, 0.26], *p* = 0.02, and that self-consistency was a significant predictor of well-being, *β* = 0.17, *SE* = 0.03, 95%CI [0.08, 0.21], *p* < 0.001. Furthermore, integration was a significant predictor of self-efficacy, *β* = 0.30, *SE* = 0.04, 95%CI [0.20, 0.39], *p* < 0.001, and that self-efficacy was a significant predictor of well-being, *β* = 0.54, *SE* = 0.03, 95%CI [0.46, 0.64], *p* < 0.001. After including mediators in the model, integration’s effect on well-being lessened, suggesting a partial mediation c = *β* = 0.35, *SE* = 0.04, 95%CI [0.26, 0.45], *p* < 0.001; c’= *β* = 0.17, *SE* = 0.04, 95%CI [0.09, 0.25], *p* = 0.001. The indirect effect was tested using a percentile bootstrap estimation approach with 5000 samples, implemented with the PROCESS Macro. These results indicated that the indirect coefficients were significant for self-consistency (*B* = 0.02, *SE* = 0.01, 95%CI [0.04, 0.70], standardized *β* = 0.02), and for self-efficacy (*B* = 0.16, *SE* = 0.03, 95%CI [0.09, 0.23], standardized *β* = 0.16). Integration was associated with well-being scores that were approximately 0.02 points higher as mediated by self-consistency, and 0.16 points higher as mediated by self-efficacy.

##### Hypothesis 2 (a, b, c, d, e)

Figure 3 presents results indicating a negative and significant total effect of compartmentalization on well-being supporting H2a. Compartmentalization was a significant negative predictor of self-consistency (*β* = −0.10, *SE* = 0.06, 95%CI [−0.20, −0.05], *p* = 0.03) and self-consistency was a significant predictor of well-being, *β* = 0.17, *SE* = 0.03, 95%CI [0.08, 0.21]. However, compartmentalization was not a significant predictor of self-efficacy (*β* = −0.07, *SE* = 0.04, 95%CI [−0.13, 0.02], *p* = 0.15), removing the possibility of its mediating effect. After including mediators in the model, the compartmentalization effect on well-being lessened, suggesting a partial mediation c = *β* = −0.20, *SE* = 0.04, 95%CI [−0.26, −0.09], *p* < 0.001; c’= *β* = −0.14, *SE* = 0.04, 95%CI [−19, −0.06], *p* = 0.001. The indirect effect was tested using a percentile bootstrap estimation approach with 5000 samples, implemented with the PROCESS Macro. These results indicated that the indirect coefficients were significant for self-consistency (*B* = 0.02, SE = 0.01, 95%CI [−0.04, −0.01], standardized *β* = −0.02) but not for self-efficacy (*B* = 0.03, SE = 0.03, 95%CI [−0.09, 0.01], standardized *β* = 0.16). Compartmentalization was associated with well-being scores that were approximately 0.02 points lower as mediated by self-consistency.

##### Hypothesis 3 (a, b, c, d, e)

Figure 4 presents the results of the third mediation model, indicating no total effect of categorization on well-being rejecting H3a. Categorization was a significant negative predictor of self-consistency (*β* = 0.16, *SE* = 0.04, 95%CI [0.06, 0.22], *p* < 0.01), and self-consistency was a significant predictor of well-being, *β* = 0.20, *SE* = 0.03, 95%CI [0.10, 0.24] *p* < 0.01. However, categorization was not a significant predictor of self-efficacy (*β* = −0.01, *SE* = 0.03, 95%CI [−0.07, 0.05], *p* = 0.86), removing the possibility of its mediating effect. After including mediators in the model, the compartmentalization effect on well-being increased and became significant c = *β* = −0.05, *SE* = 0.03, 95%CI [−0.12, −0.01] *p* < 0.001; c’ = *β* = −0.10, *SE* = 0.04, 95%CI [−0.13, −0.01], *p* = 0.001. The indirect effect was tested using a percentile bootstrap estimation approach with 5000 samples, implemented with the PROCESS Macro. These results indicated that the indirect coefficients were significant for self-consistency (*B* = 0.02, *SE* = 0.01, 95%CI [0.01, 0.06], standardized *β* = 0.03) but not for self-efficacy (*B* = −0.01, *SE* = 0.03, 95%CI [−0.06, 0.04], standardized *β* = −0.01). It might be inferred that for categorization, the total effect on well-being was insignificant, but the direct was significant and negative. Such a situation is implied when opposite effects result from the independent variable. In the present model, categorization, on one side, enhanced internal consistency but, on the other, decreased well-being via different mechanisms. The hypothesis that it might be via a decrease in self-efficacy was rejected.

## 4. Discussion

### 4.1. Associations between Multicultural Identity Configurations, Self-Consistency, Self-Efficacy, and Well-Being

The correlational analyses identified some significant pairwise associations between multicultural identity configurations, self-concept consistency and efficacy, and well-being. Hence, the present study has expanded existing knowledge linking cultural identity with self-concept (i.e., [47,63]) by including a more complex multicultural identity paradigm (reflective of social changes in the global era). However, not all variables were significantly associated. Possible reasons are discussed in the following sections.

Out of multicultural identity configurations, the strongest positive correlation was found between integration and self-efficacy. Integration was also a positive correlate of self-consistency. It supported previous suggestions that accepting all cultural paradigms within oneself can help boost competencies, leading to better adjustment [64,65]. Our findings also extend the prior assumption of self-efficacy acting as a buffer to experienced cultural homelessness and complexity while supporting integration [8], because both variables were also positively associated with well-being. Our data further evidenced that multicultural identity categorization was positively associated with self-consistency but not self-efficacy. It might be that essentialist and categorical strategies, which are the bases of identity categorization [3], increase motivation for a sense of self-concept consistency for individuals navigating different cultural frames. However, in contrast to integration, categorization implies the suppression of parts of oneself to achieve internal consistency. Categorization and compartmentalization of multicultural identity may lead to self-discriminant attitudes similar to categorical attitudes towards culturally diverse “others”, causing discriminative behaviors towards them [66]. Self-discriminant processes, in turn, reduce TCKs’ cross-cultural abilities and hence impair their self-efficacy. For our sample, the associations between categorization and compartmentalization with self-efficacy were negative but at borderline significance. For TCKs in the UAE, both exclusive configurations might not seem relevant to their efficacy. It might be related to the country’s integrative and inclusive efforts and policies. However, more studies are needed to expand upon this. Importantly, an integrational strategy of dealing with multicultural identity constitutes a more sustainable alternative to categorization and compartmentalization as it seems supportive of efficacy. Finally, in line with earlier research [3], compartmentalization of multicultural identity was negatively associated with self-consistency. Such findings validate earlier qualitative studies on TCKs’ identity fragmentation [8,38].

Regarding well-being associations with multicultural identity configurations, integration was positively, and compartmentalization was negatively associated, though categorization was not connected considerably. Such evidence validates earlier claims that multicultural identity constructs are related to well-being [3,67]. Our findings reconcile contradictory studies on the well-being of multicultural individuals raised between cultures [8,55], implying that configurations of multicultural identities moderate whether TCKs would function well. It is not the mere exposure to diversity but the internal integration versus identity compartmentalization that matters to the well-being of TCKs. Hence, the present study extends previous claims proposing integration as a crucial and enhancing factor in dealing with multiculturalism [3,42]. Additionally, the positive association of self-efficacy with well-being supported previous findings [57], linking positive beliefs about one’s abilities with better functioning and adaptive strategies for multicultural individuals. Furthermore, self-consistency was positively related to well-being. The literature explains that a satisfied sense of self-consistency, central to personal identity, may be associated with positive emotions and enhanced well-being [23,56].

### 4.2. Mediation Models Interpretation

To explain the direct effect of integration and compartmentalization on well-being, we have included self-consistency and self-efficacy as possible mediators suggested in the existing literature on TCKs’ functioning [38,68]. Our models revealed significant intermediary effects in multicultural identity configurations’ impacts on well-being. 

Integration of multicultural identity supported well-being directly, as previously explained, and indirectly through its positive impact on self-consistency and self-efficacy. Integration may support the creation of a hybrid identity encompassing unique cultural composition making one maintain a unique but consistent self-concept. We conclude that integrating all cultural paradigms into oneself stimulates cross-cultural competencies supportive of self-efficacy and creates more sense of self-concept consistency and hence positive self-views that increase life satisfaction and functioning. Such inferences confirm previously suggested processes [68]. Conversely, if TCKs structure their identity based on cultural fragmentation (compartmentalization), their identity consistency can decline, negatively impacting on well-being [8,55].

Interestingly, though no total effect was revealed between categorization and well-being, the mediational model implied an indirect negative effect. Such a situation occurs when there are two opposing causal pathways from a predictive variable. Categorization entails, on one side, increased internal consistency but, at the same time, decreased well-being. Our hypothesis suggesting that it might happen via impaired self-efficacy was rejected. We suggest that to fulfil a motive for self-consistency, one suppresses parts of oneself (categorized multicultural identity), which further weakens the ability to adjust and hence decreases well-being. Another explanation could be that the roots of the categorical strategies lie in rigidity and an essentialist mindset [3], which reduces inclusion and acceptance of one’s diverse cultural frames, negatively impacting self-attitudes. Hence, categorization as a configuration of multicultural identity may lead to self-discriminant attitudes similar to categorical attitudes towards culturally diverse “others”. Self-discriminant processes, in turn, reduce TCKs’ cross-cultural abilities. However, more exploratory studies with other possible mediators are needed to verify these hypotheses.

In summary, the mediation models highlighted the prominent role of cultural integration in third culture individuals’ well-being and pointed to its supportive role in forming self-consistency and self-efficacy. We also explained the mechanism behind the positive changes brought by multicultural identity integration suggested in the existing literature [3,68]. Previous studies indicated that multiculturalism stemming from integration and acceptance, as opposed to exclusion-based assimilation, has created a favorable social context and increased self-esteem, leading to boosted well-being [69]. Our study expanded such claims to the internal cultural diversity of third culture individuals.

### 4.3. Limitations and Future Directions

Despite its significance, the present study has some limitations. Firstly, the sample had an unequal gender distribution, and the participants were from diverse origins, with a prevalence of South Asian individuals. The potential influence of their own cultural backgrounds on our results cannot be excluded, limiting our study’s generalizability. Nevertheless, our participants were of less-studied Eastern origin, and hence, the present study assists in a better understanding of TCKs outside the Westernized perspective. Another limitation concerns the identification of the participants as TCKs, assumed based on a single definition, which may seem limited. Further studies on the level of identification with TCKs and well-being variables are hence recommended. Additionally, there has been some criticism regarding the possibility of people with similar life experiences constituting a social category or a “culture” [70]. However, whilst most literature on the topic relies on the self-identification of adult TCKs, this is still only a quasi-social category. Hence, the results of the present evaluation may also apply to participants categorized more broadly as bicultural, multicultural, or sojourners.

Furthermore, the mediational model and cross-sectional character of the research have indicated limitations related to dependence on initial hypotheses and the inability to conclude actual causal relationships or long-term associations. The mediation effects were weak, suggesting the existence of possible other intermediary variables that could explain our model further. In particular, seeking other mediators of the categorization’s effects on well-being as inferred from our analyses’ outcome is recommended. Categorization implies exclusion, a rigid mindset, and essentialistic tendencies [3]. Therefore, we propose exploring the factors of mindset rigidity or psychological flexibility as possible mediators.

## 5. Conclusions

Internationalizations of economies and ease in mobility have increased the numbers of children who grow up between cultures in “mobile” families around the globe. In the UAE, youths raised exposed to cultures different than their parent(s) constitute a majority. Therefore, the research on factors supporting their function is highly salient. This study offered four main contributions. Firstly, we expanded knowledge on TCKs living in a specific context of the multicultural United Arab Emirates. While other studies in the UAE focused primarily on the adverse effect of TCKs’ confused multicultural identity on mental health [11], our study explored factors supportive of TCKs well-being which may be used as directions for interventions facilitating TCKs’ functioning. In particular, the integration of cultural paradigms seems relevant. Hence, we add to a slowly growing literature on TCKs in the UAE [3,11,12]. Furthermore, our findings suggest that not mere exposure to diversity but internal integration versus identity compartmentalization moderate the well-being of TCKs. Hence, our study contributed to a better understanding of the TCKs’ identity paradigm and pointed to multicultural identity integration as vital to the positive functioning of TCKs. Thirdly, we linked the multicultural identity configurations with aspects of self-concept, namely self-consistency and self-efficacy. Lastly, we explored the mechanism behind the multicultural identity configurations’ effect on well-being, expanding previous research in this area. The study pointed to the mediating roles of self-consistency and self-efficacy in enhancing the effect of multicultural identity integration on well-being. Alternatively, TCKs with a more compartmentalized multicultural identity might have decreased well-being partially because of a reduced sense of self-consistency. Accounting for increasing numbers of multicultural individuals with transient lifestyles worldwide, this research outcome is valuable to social sciences concerned with the health and functioning of future generations.

## Figures and Tables

**Figure 1 ijerph-20-03880-f001:**
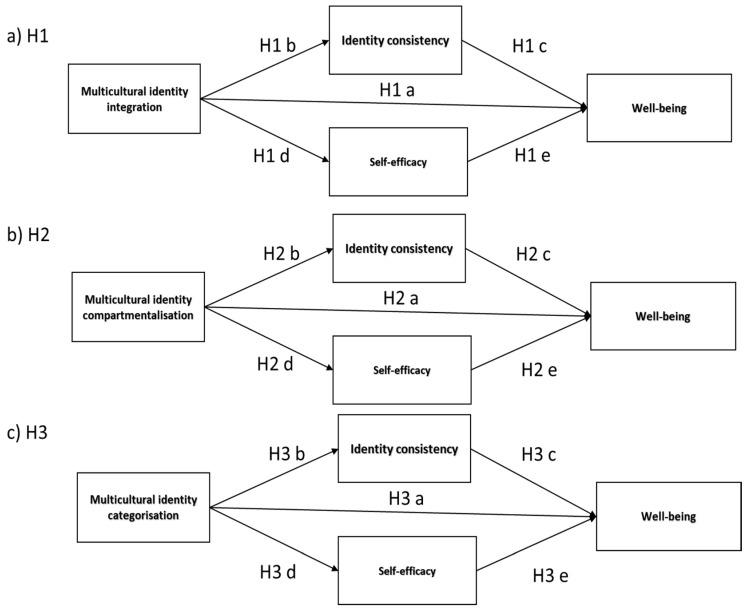
Hypothesized mediation effects of self-consistency and self-efficacy in a predictive impact of multicultural identity configurations (**a**) integration (H1), (**b**) compartmentalization (H2), (**c**) categorization (H3) on well-being (straight line—direct effects; dashed line—mediating effect).

**Figure 2 ijerph-20-03880-f002:**
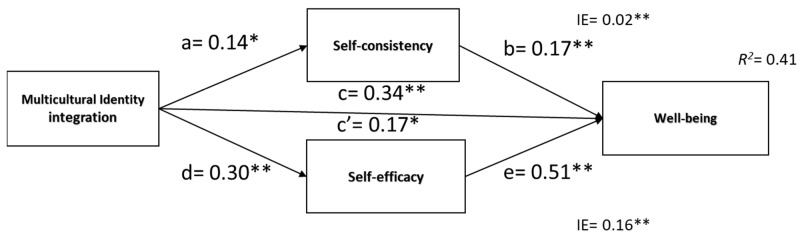
Mediation model of indirect effects of self-consistency and self-efficacy in the predictive effect of multicultural identity integration on well-being. Note: ** *p* < 0.001,* *p* < 0.05.

**Figure 3 ijerph-20-03880-f003:**
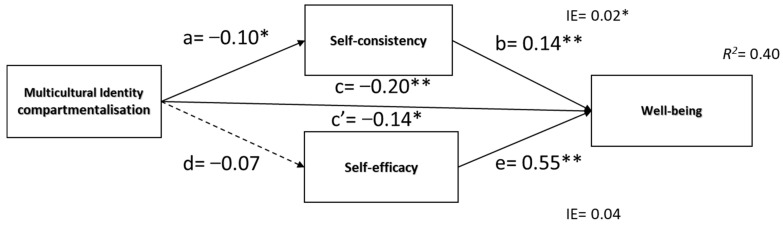
Mediation model of indirect effects of self-consistency and self-efficacy in the predictive effect of multicultural identity compartmentalization on well-being. Note: ** *p* < 0.001,* *p* < 0.05.

**Figure 4 ijerph-20-03880-f004:**
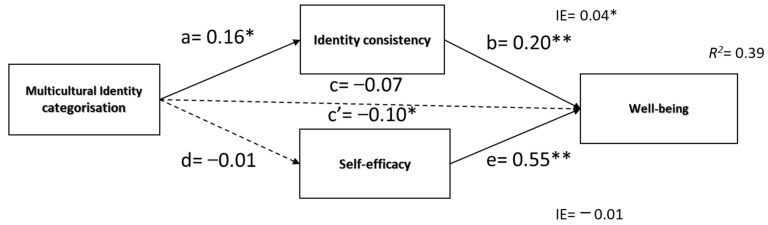
Mediation model of indirect effects of self-consistency and self-efficacy in the predictive effect of multicultural identity categorization on well-being. Note: ** *p* < 0.001,* *p* < 0.05.

**Table 1 ijerph-20-03880-t001:** Descriptive statistics and pairwise correlations.

Variables (*n* = 399)	*M (SD)*	1	2	3	4	5	6
1. Self-consistency	3.88 (1.25)	-					
2. Self-efficacy	5.00 (0.98)	**16 ***	-				
3. Integration	4.91 (1.02)	**0.11 ***	**0.31 ****	-			
4. Categorization	4.19 (1.34)	**0.16 ***	−0.07	**−0.17 ****	-		
5. Compartmentalization	3.88 (1.09)	**−0.10 ***	−0.08	0.05	**0.48 ****	-	
6. Well-being	4.48 (0.74)	**0.23 ****	**0.48 ****	**0.22 ****	−0.07	**−0.29 ****	-

Note: * *p* < 0.05; ** *p* < 0.001. Significant correlations are presented in bold.

**Table 2 ijerph-20-03880-t002:** Hierarchical regression analysis of multicultural identity configurations, self-consistency and self-efficacy on well-being.

Predictors	Model 1			Model 2		
	B	SE	Beta	B	SE	Beta
Integration	0.25	0.04	**0.28 ****	0.09	0.04	**0.11 ***
Compartmentalization	−0.20	0.04	**−0.23 ****	−0.12	0.03	**−0.14 ****
Categorization	0.02	0.03	−0.07	−0.03	0.03	−0.07
Self-consistency				0.15	0.03	**0.18 ****
Self-efficacy				0.56	0.04	**0.52 ****
R sqr (R sqr Adj)	0.35 (0.11)			0.65 (0.41)		
F	**18.00 (3; 395) ****			**36.98 (5; 393) ****		
Delta R sqr	**0.12 ****			**0.30 ****		

Note: ** *p* < 0.001,* *p* < 0.05. Significant effects are presented in bold.

## Data Availability

The data presented in this study are available on request from the corresponding author.

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
