# Peer review of "Multicultural Identity Integration versus Compartmentalization as Predictors of Subjective Well-Being for Third Culture Kids: The Mediational Role of Self-Concept Consistency and Self-Efficacy"

_ijerph, 2023, doi:10.3390/ijerph20053880_

Round 1

Reviewer 1 Report

The article is well-written, and the topic is relevant. Several variables are considered for understanding the well-being of TCK kids, while reading the article I found some issues that should be considered by the authors before publication of the article.

Introduction

In the third hypothesis please indicate what do you mean by “changes in self-consistency and self-efficacy”, and how change is measured in that context.

Methods

I think it will be useful for the readers to have more information about the scales used, specifically the psychometric properties and where have bee those properties explored. Probably, it will be relevant to note other papers using those scales with similar populations.

Please check these two sentences, I understand that you conducted the CFA analysis and got those results, if that is the case, please rewrite those sentences, to make it clearer.

“Confirmatory Factor Analysis (CFA) to assess the three-factor structure of the MULTIIS. The CFA model provided acceptable fit to the data: χ2 = 277 398,41; df = 196; CMIN/df = 2,03; RMSEA = .051 [90 % CI = .044–.060]; CFI = .924”

Please specify how you approached to the sample to understand the anonymity and confidentiality.

Results

Please clarify whether this sentence is related to well-being instead of life satisfaction, I think is easier to follow if the name of the construct is used consistently.

Discussion

In the first paragraph of the discussion, I think that although there are significant correlations between the mentioned constructs, those correlations are low, and probably you need to discuss that. Also, not all the correlations are significant.

In the limitations subsection you suggest possible other mediators for categorization effects, could you suggest which would be those factors based on your research and knowledge.

“In particular, seeking other mediators of the categorisation’s effects on well-being as inferred from our analyses' outcome is recommended.”

Also, as a limitation you might consider how you identified TCKs, it was only one question. It will be nice to indicate alternative options that might enhance the validity of the identification of that population in future studies. 

Author Response

Comments and Suggestions for Authors

The article is well-written, and the topic is relevant. Several variables are considered for understanding the well-being of TCK kids, while reading the article I found some issues that should be considered by the authors before publication of the article.

Introduction

In the third hypothesis please indicate what do you mean by “changes in self-consistency and self-efficacy”, and how change is measured in that context.

Answer: Dear Reviewer, thank you for spotting this inconsistency. We have now adjusted the hypothesis to express the mediation analysis more adequately.

H3. Multicultural identity categorisation predicts negatively well-being (H3a), and such an effect is mediated by the level of self-consistency (H3b,c) and self-efficacy (H3d,e).

Methods

I think it will be useful for the readers to have more information about the scales used, specifically the psychometric properties and where have bee those properties explored. Probably, it will be relevant to note other papers using those scales with similar populations.

We have now added the suggested information about the scales’s properties in studies with similar populations

The GSES has shown reliability in past studies on predictors of life satisfaction with students of non-Western origin (e.g., Capri et al., 2012).

SCS has demonstrated reliability in cross-cultural studies (Vignoles et al.,2016).

The MULTIIS scale has been employed in an exploratory study on female TCKs (Mosanya & Kwiatkowska, 2021) and has shown a three-factor structure and reliability of subscales.

The scale showed reliability in the cross-cultural assessment of adolescents (Grob et al. 1996).

Please check these two sentences, I understand that you conducted the CFA analysis and got those results, if that is the case, please rewrite those sentences, to make it clearer.

“Confirmatory Factor Analysis (CFA) to assess the three-factor structure of the MULTIIS. The CFA model provided acceptable fit to the data: χ2 = 277 398,41; df = 196; CMIN/df = 2,03; RMSEA = .051 [90 % CI = .044–.060]; CFI = .924”

Answer: The sentences have now been rewritten:

Confirmatory Factor Analysis (CFA) was performed to assess the three-factor structure of the MULTIIS. The CFA model provided acceptable fit to the data: χ2 = 398,41; df = 196; CMIN/df = 2,03; RMSEA = .051 [90 % CI = .044–.060]; CFI = .924.

Please specify how you approached to the sample to understand the anonymity and confidentiality.

Answer: Thank you for this comment. We have added the requested information in the procedure section.

To assure anonymity and data confidentiality, the link to the study was posted on groups and platforms for international students in the UAE. The contribution was voluntary, and participants were given an email to the researchers and asked to insert their initials only. This ought to be mentioned in any communication should they wish to withdraw from the study. The data was encrypted for safe storage.  

Results

Please clarify whether this sentence is related to well-being instead of life satisfaction, I think is easier to follow if the name of the construct is used consistently.

Answer: Thank you for pointing out this error. We have now changed the wording to well-being for consistency.

Discussion

In the first paragraph of the discussion, I think that although there are significant correlations between the mentioned constructs, those correlations are low, and probably you need to discuss that. Also, not all the correlations are significant.

Answer: Thank you for this point. We have now rewritten the paragraph accounting for your suggestions.

The correlational analyses identified some significant pairwise associations between multicultural identity configurations, self-concept consistency and efficacy, and well-being. Hence, the present study has expanded existing knowledge linking cultural identity with self-concept (i.e. 45; 59) by including a more complex (reflective of social changes in the global era) multicultural identity paradigm. However, not all variables were significantly associated. Possible reasons are discussed in the following sections.

Out of multicultural identity configurations, the strongest positive correlation was found between integration and self-efficacy. Integration was also a positive correlate of self-consistency. It supported previous suggestions that accepting all cultural paradigms within oneself can help boost competencies, leading to better adjustment (60; 61). Our findings also extend the prior assumption of self-efficacy acting as a buffer to experienced cultural homelessness and complexity while supporting integration (8) because both variables were also positively associated with well-being. Our data further evidenced that multicultural identity categorisation was positively associated with self-consistency but not self-efficacy. It might be that essentialist and categorical strategies, which are the bases of identity categorisation (3), increase motivation for a sense of self-concept consistency for individuals navigating different cultural frames. However, in contrast to integration, categorisation implies the suppression of parts of oneself to achieve internal consistency. Categorisation and compartmentalisation of multicultural identity may lead to self-discriminant attitudes similarly to categorical attitudes towards culturally diverse "others", causing discriminative behaviours towards them (Bastian & Haslam, 2006). Self-discriminant processes, in turn, reduce TCKs' cross-cultural abilities and hence impair their self-efficacy. For our sample, the associations between categorisation and compartmentalisation with self-efficacy were negative but at borderline significance. For TCKs in the UAE, both exclusive configurations might not seem primarily relevant to their efficacy. It might be related to the country's integrative and inclusive efforts and policies, but more studies are needed. Importantly,, an integrational strategy of dealing with multicultural identity constitutes a more sustainable alternative to categorisation and compartmentalisation as it seems supportive of efficacy. Finally, in line with earlier research (3), the compartmentalisation of multicultural identity was revealed to be negatively associated with self-consistency. Such findings validate earlier qualitative recalls on TCKs' identity fragmentation (8; 36).

Regarding well-being associations with multicultural identity configurations, integration was positively, and compartmentalisation was negatively associated, though categorisation was not connected considerably. Such evidence validates earlier claims that multicultural identity constructs are significantly related to well-being (62; 3). Our findings reconcile contradictory studies on the well-being of multicultural individuals raised between cultures (8; 53), implying that configurations of multicultural identities moderate whether TCKs would function well. It is not the mere exposure to diversity but the internal integration versus identity compartmentalisation that matters to the well-being of TCKs. Hence, the present study extends previous claims proposing integration as a crucial and enhancing factor in dealing with multiculturalism (40; 3). Additionally, the affirmative associations of self-efficacy with well-being supported previous findings (55), linking positive beliefs about one's abilities with better functioning and adaptive strategies for multicultural individuals. Furthermore, self-consistency was positively related to well-being. The literature explains that a satisfied sense of self-consistency, central to personal identity, may be associated with positive emotions and enhanced well-being (54; 22).

In the limitations subsection you suggest possible other mediators for categorization effects, could you suggest which would be those factors based on your research and knowledge.

“In particular, seeking other mediators of the categorisation’s effects on well-being as inferred from our analyses' outcome is recommended.”

Answer: We have added the possible factors.

In particular, seeking other mediators of the categorisation’s effects on well-being as inferred from our analyses' outcome is recommended. Categorisation implies exclusion, rigid mindset and essentialistic tendencies (Mosanya & Kwiatkowska, 2021). Therefore we propose an exploration of the factors of a mindset rigidity or psychological flexibility as possible mediators.

Also, as a limitation you might consider how you identified TCKs, it was only one question. It will be nice to indicate alternative options that might enhance the validity of the identification of that population in future studies. 

Answer: Thank you for this suggestion. We have now included this limitation in our paper.

Another limitation concerns the identification of the participants as TCKs, assumed based on single definition, which may seem limited. It may seem limited. Further studies on the level of identification with TCKs and well-being variables are hence recommended. Additionally, there has been some critical voice against the possibility of individuals with similar life experiences constituting a social category or a “culture” (Pearce, 2011).  Still, most literature on the topic relies on the self-identification of adult TCKs, this is still only a quasi-social category. Hence, the results of the present evaluation may apply also to participants categorised more broadly as biculturals, multicultural or sojourners

Reviewer 2 Report

I think that this paper is well-written. The narrative is well-structured and the arguments are clearly presented. I think this paper is worth being published in the journal. 

However, I would like to make some suggestions to improve the paper before its publication:

1 - A small technical remark. Please check the font size. Some sentences and paragraphs (e.g. lines 117-118, 138-142, Conclusion, etc.) have a different font size, at least in the pdf-generated file of your article. 

2 - I think that the introduction is too long. It might be useful to differentiate between Introduction and Literature Overview.

3 - I think that you could expand your conclusion, by adding more details about how your research has contributed to the academic research on TCK in the UAE. It is interesting to learn more about the state of research on TCK-related issues in the UAE (mention local authors, papers, research projects, and research funding, if any).  

Author Response

Comments and Suggestions for Authors

I think that this paper is well-written. The narrative is well-structured and the arguments are clearly presented. I think this paper is worth being published in the journal. 

Answer: Thank you for your kind and supportive words.

However, I would like to make some suggestions to improve the paper before its publication:

1 - A small technical remark. Please check the font size. Some sentences and paragraphs (e.g. lines 117-118, 138-142, Conclusion, etc.) have a different font size, at least in the pdf-generated file of your article. 

Answer: Thank you for this remark. We have now adjusted and uniformised the format of our work.

2 - I think that the introduction is too long. It might be useful to differentiate between Introduction and Literature Overview.

Answer: As per Reviewer’s suggestion we have now change the headings and differentiated between the Introduction and the Literature Overview. The first introductory paragraph has been now titled Introduction while the following sections are parts of Literature review.

3 - I think that you could expand your conclusion, by adding more details about how your research has contributed to the academic research on TCK in the UAE. It is interesting to learn more about the state of research on TCK-related issues in the UAE (mention local authors, papers, research projects, and research funding, if any).

Answer: We have included suggested details as below:

Internationalisations of economies and ease in mobility increased the numbers of children who grow up between the cultures in “mobile” families around the globe. In the UAE youths raised exposed to cultures different than their parent(s) constitute majority. Therefore, the research on factors supporting their function are highly salient. This study offered four main contributions. Firstly, we expanded knowledge on TCKs living in a specific context of multicultural United Arab Emirates. While other studies done in the UAE focused primarily on negative effects of confused multicultural identity of TCKs on mental health (Habeeb, 2019), our study provided an exploration of the factors supportive of TCKs well-being which may be used as directions for interventions facilitating TCKs’ functioning. In particular the integration of the cultural paradigms seems relevant. Hence, we add to a slowly growing literature on TCKs in the UAE (Dillon & Ali, 2019; Habeeb &Hameed, 2021; Mosanya & Kwiatkowska, 2021).